# Exploring pain interference with motor skill learning in humans: A systematic review

**David Matthews** [1] *, **Edith Elgueta Cancino** [1], **Deborah Falla** [1], **Ali Khatibi** [1,2]

**1** Centre of Precision Rehabilitation for Spinal Pain (CPR Spine), School of Sport, Exercise and Rehabilitation Sciences, University of Birmingham, Birmingham, United Kingdom, **2** Centre for Human Brain Health, University of Birmingham, Birmingham, United Kingdom

* DXM986@student.bham.ac.uk

## Abstract

Motor learning underpins successful motor skill acquisition. Although it is well known that pain changes the way we move, it's impact on motor learning is less clear. The aim of this systematic review was to synthesize evidence on the impact of experimental and clinical pain on task performance and activity-dependent plasticity measures across learning and explore these findings in relation to different pain and motor learning paradigms. Five databases were searched: Web of Science, Scopus, MEDLINE, Embase and CINAHL. Two reviewers independently screened the studies, extracted data, and assessed risk of bias using the Cochrane ROB2 and ROBIN-I. The overall strength of evidence was rated using the GRADE guidelines. Due to the heterogeneity of study methodologies a narrative synthesis was employed. Twenty studies were included in the review: fifteen experimental pain and five clinical pain studies, covering multiple motor paradigms. GRADE scores for all outcome measures suggested limited confidence in the reported effect for experimental pain and clinical pain, on motor learning. There was no impact of pain on any of the task performance measures following acquisition except for 'accuracy' during a tongue protrusion visuomotor task and 'timing of errors' during a motor adaptation locomotion task. Task performance measures at retention, and activity dependent measures at both acquisition and retention showed conflicting results. This review delivers a detailed synthesis of research studies exploring the impact of pain on motor learning. This is despite the challenges provided by the heterogeneity of motor learning paradigms, outcome measures and pain paradigms employed in these studies. The results highlight important questions for further research with the goal of strengthening the confidence of findings in this area.

## 1. Introduction

Motor skill learning is fundamental to life. Such learning allows individuals to adapt to their 'ever-changing' environment and engage in new challenges. Motor learning is characterised as repeated task practice resulting in efficient identification, selection [1] and effortless performance of movement [2]. Motor learning can be divided into two stages, characterised by the rate and timing of learning: early or acquisition (single session, online learning) and late or consolidation (multiple sessions/time periods). Research has utilised the above delineation of motor learning to explore neural networks underlying these stages.

**Data Availability Statement:** All relevant data are within the paper and its Supporting Information files.

**Funding:** The author(s) received no specific funding for this work.

**Competing interests:** The authors have declared that no competing interests exist.

Repeated practice is fundamental to both acquisition and maintenance of learning. It aids the transition from a vulnerable to a stable state memory which in turn increases resistance to interference. Brasher-Krug et al. [3] were the first to demonstrate interference with motor learning. The authors demonstrate how practicing a second motor skill attenuated consolidation (offline learning) of a previously acquired skill. This type of interference has been referred to as retrograde interference and demonstrated the vulnerability of consolidation [4]. Subsequent studies have demonstrated declarative learning tasks [5] and non-invasive brain stimulation [6] can cause retrograde interference of motor learning. Conversely, anterograde interference describes the process whereby learning of an initial task, impacts on the acquisition of a second task [7], suggesting that acquisition and associated neuroplastic changes may be influenced by history or current state. Research also suggests interference of motor skill acquisition with age [8], emotion [9], non-invasive brain stimulation [6] and pain [10].

Pain as a sensory and emotional experience is unpleasant, demands attention [11], and can disrupt task related goals [12]. Motor skill learning is widely utilised in rehabilitation both in the presence and absence of pain [13, 14]. This is despite Boudreau et al. [15] suggesting in a narrative review that to optimize success, motor skill learning should be performed in a pain free manner. More recent evidence on the interaction between pain and motor skill learning [16] has questioned this view point. Therefore, understanding how pain interferes with motor learning may help to plan and implement motor skill learning interventions.

The eclectic nature of motor skill learning paradigms used within research complicates the goal of finding consensus in the literature. Research suggests that underlying neural mechanisms [17] and vulnerability to interference may depend on the motor learning paradigm [18]. In addition, different pain paradigms, such as tonic, phasic or clinical pain [19], or the tissue stimulated [20, 21] could potentially alter the corticospinal activation occurring as a result of the task.

Two systematic reviews [22, 23] have been recently published in this area. Izadi et al. [22] investigated the impact of acute experimental pain on behavioural performance following motor learning. The authors concluded that the majority of studies reported negative impacts of acute pain on motor learning. Results were grouped based on the tissue where the pain was induced and investigated only experimental tonic pain paradigms. Stanisic et al. [23] explored the impact of experimental and clinical pain on motor learning induced neuroplasticity, assessed using TMS. They reported both acute and clinical pain may impede neuroplasticity, resulting from motor learning, while highlighting a diversity of study findings. Neither of the above systematic reviews differentiated the impact of different motor learning paradigms on this interaction despite identifying that this may influence the findings.

The objective of this research is to systematically review, critically evaluate methods and summarise the present literature examining the impact of pain on task performance measures and measures of neuroplasticity associated with the cerebellum and corticospinal tract, including the primary motor cortex, following motor skill learning. This is the first systematic review to delineate the impact of experimental and clinical pain on a multitude of different behavioural and neuroplasticity outcomes measures and looks to compare and contrast these findings related to different pain paradigms (experimental tonic or phasic pain and clinical pain) and different motor paradigms (motor sequence learning, visuomotor learning, repeated ballistic movements, motor adaptation and ecological learning).

## 2. Methods

This systematic review followed a pre-defined published protocol [24] and is reported in concordance with the Preferred Reporting Items for Systematic Review and Meta-Analysis

(PRISMA) [25, 26]. The protocol was registered on PROSPERO (CRD42020213240) on 15th October 2020. The methodology outlined below did not change from the published protocol.

## 2.1. Eligibility criteria

The eligibility criteria followed the PICOS framework and will be discussed briefly here; further details on inclusion and exclusion of studies are described in the protocol [24].

## 2.2. Inclusion criteria

**2.2.1. Populations.** Studies were eligible for inclusion in this review if they included an experimental group consisting of adults (age $\geq$ 18 years old) who experienced either experimental pain (tonic (long lasting) or phasic (brief noxious stimuli)) or clinical pain. Clinical pain was considered as any symptom of pain included in the IASP definition for pain [27] excluding those occurring in the presence of neurological disease or due to delayed onset of muscle soreness.

**2.2.2. Intervention.** Motor skill learning interventions were selected using the definitions presented in Table 1. This encompassed the most common motor paradigms used in research including both, implicit and explicit: motor sequence learning, visuomotor learning, repeated ballistic movements, motor adaptation and ecological learning. Prism adaptation paradigms were excluded from this review in an attempt to reduce confounding variables, such as the impact of visual perception.

**2.2.3. Comparator.** A comparison group consisting of adults (age$\geq$ 18 years old) with no pain was required for inclusion. All subjects were required to complete the same motor skill learning to compare learning in the presence of pain versus learning in the absence of pain.

**2.2.4. Outcomes.** Primary outcome measures of interest in this review included task performance measures related to motor learning, or activity-dependent plasticity measures (See protocol for more details). Measures of task performance included both spatial and temporal measures. Activity dependent plasticity measures included, but not limited to, changes in amplitude, temporal or spatial characteristic obtained using magnetic resonance imaging (MRI), transcranial magnetic stimulation (TMS) or electroencephalogram (EEG).

**2.2.5. Study design.** Randomised control studies (RCTs) and non-randomized studies were included. Single case studies, case series and review papers along with any studies not published in English were excluded. Also excluded were any study including treatments as an adjunct to motor learning, in an attempt to minimise heterogeneity of interventions subjects were exposed to and consequently minimise confounding variables that may impact on the amount of learning observed. Study duration was not limited.

**Table 1. Classification of motor learning.**

| |
|---|
| Motor sequence learning: 'refers to the process by which simple, well defined movement elements come to be performed effortlessly as a unitary sequence through repeated practice' [2]. |
| Motor adaptation: 'involves adjusting how an already well-practiced action is executed to maintain performance in response to a change in the environment (e.g. force, visual etc) or the body' [1]. |
| Repeated ballistic movements: repeated brisk movements of a single joint. |
| Visuomotor learning: capacity to identify and perform novel movements of a visually guided motor task efficiently and effortlessly through repeated practice. |
| Ecological learning: refers to performance of movements that reflect real-world tasks through repeated practice. |

## 2.3. Information sources

Comprehensive searches of the following databases were completed from inception until July 2022: Web of Science, Scopus, MEDLINE, Embase and CINAHL. In addition, hand searching of preprint repositories, including PsyArxiv and BioArxiv was completed.

## 2.4. Search strategy

Search strategies were designed with assistance from a subject librarian, including MeSH terms and natural language combinations and adapted for use with above databases.

## 2.5. Data management

Articles resulting from the search process were downloaded to Endnote (V9 or later) software (Clarivate Analytics) and duplicates identified and deleted.

## 2.6. Study selection

Two reviewers (DM and EEC) independently screened titles and abstracts against the predetermined inclusion and exclusion criteria. Full articles were downloaded for those provisionally included. In circumstances where information on screening data was not present authors were contacted by e-mail. A failure to respond after four weeks including a reminder e-mail at two weeks resulted in the study being excluded. Once the above procedure had been completed and full texts collated, the screening process was repeated. Information on, and reasons for excluding studies was reported. Any disagreements were discussed, and a third reviewer (AK) was consulted as required.

## 2.7. Data extraction

Data extraction was performed independently by two reviewers (DM and EEC) using a data extraction form developed from information gathered from early literature scoping activities and piloted before use. Authors were contacted if clarity was required using the process outlined above.

## 2.8. Risk of bias

The Cochrane risk of bias tool (ROB2) and the risk of bias in non-randomized studies of interventions (ROBINS-I) was used to assess risk of bias for RCTs and non-randomized studies respectively. Risk of bias for each outcome measure, for each domain was independently assessed by the two reviewers (DM and EEC) using the appropriate tool and an overall risk of bias judgement recorded for each measure. The decision to complete risk of bias judgements for each outcome measure within a study rather than on the entire study was based on the differing characteristics of outcome measures and the impact this had on judgements for domain 4 and 6 of ROB2 and ROBIN-I respectively. For example, 'number of errors' and 'speed' were commonly assessed jointly in motor sequence learning studies. Similar to previous studies [28], participants 'number of errors' reached learning saturation quickly; introducing bias. A further example was whether studies collected data on sleep. Sleep has the potential to impair consolidation of motor learning [2] therefore impacting retention but not acquisition. The Cohen Kappa coefficient was calculated to explore agreement between the two reviewers. Any disagreements were discussed and where appropriate a third reviewer (AK) was consulted.

## 2.9. Data synthesis

Due to the heterogeneity of motor skill learning paradigms, outcome measures, pain paradigms and a lack of reported effect sizes or variance, a quantitative synthesis and meta-analysis was not feasible. As a meta-analysis was not judged to be possible, statistical heterogeneity was not deemed appropriate. P-values establishing a significance difference between means between groups was not consistently reported in the 18 studies and therefore a narrative synthesis was employed.

## 2.10. Confidence in cumulative evidence

To aid the communication of the results of this systematic review the Grading of Recommendations Assessment, Development, and Evaluation (GRADE) approach was utilised for each outcome measure [29]. GRADE was utilised to assess the quality of evidence for each outcome measure to provide a judgement on certainty of evidence to support further recommendations. Effect sizes or sufficient data to calculate effect sizes was unavailable in all but one of the studies and was therefore not included in the analysis. Details of the studies were initially grouped under headings for the three pain paradigms, experimental tonic pain, experimental phasic pain, and clinical pain. Under each heading the five motor skill learning domains: motor sequence learning, visuomotor learning, repeated ballistic movements, motor adaptation and ecological learning, were discussed. Subsequent analysis of individual outcome measures, risk of bias and certainty of evidence was presented.

## 3. Results

### 3.1. Characteristics of studies

In total 1527 articles were identified from the searches and 742 of these were screened following removal of duplicates. 20 articles were included in the narrative synthesis. Details of the screening process, including reasons for exclusion, are detailed in Fig 1. Characteristics of the studies and findings are summarised in Tables 2 and 3 respectively. Fifteen studies were RCTs, exploring the impact of experimental pain on motor learning using between subject (thirteen) and within subject (two) designs and five were non-randomized control studies utilising clinical pain cohorts. All experimental pain studies used tonic pain paradigms (one thermal, ten capsaicin and four hypertonic saline injection). Out of the 15 studies, pain was located at the wrist/forearm in eight, upper arm in two, lower leg in three, neck in one and tongue in one. No study investigated the impact of phasic experimental pain on motor learning. The populations studied in the five clinical pain studies included osteoarthritis (OA) of the thumb, subclinical neck pain (SCNP), clinical neck pain (CNP), isolated ankle pathology and chronic tension type headaches (CTTH).

   All five categories of motor learning discussed in the methodology were represented in the studies. Four studies used motor sequence learning, six visuomotor learning, three repeated ballistic movements, six motor adaptation and one ecological learning. All studies presented at least one task performance outcome measure and 10 of the 20 studies also presented an activity dependent plasticity measure. Out of the 10 studies, four studies presented data from EEG and six presented data on TMS measures. The four studies utilising EEG collected data on the amplitude (mV) of the early somatosensory evoked potentials (SEPs) peaks only. All six of the TMS studies collected single pulse data and two collected paired pulse data. No experimental or clinical pain study explored blood oxygenation level dependent (BOLD) signal measures collected with MRI.

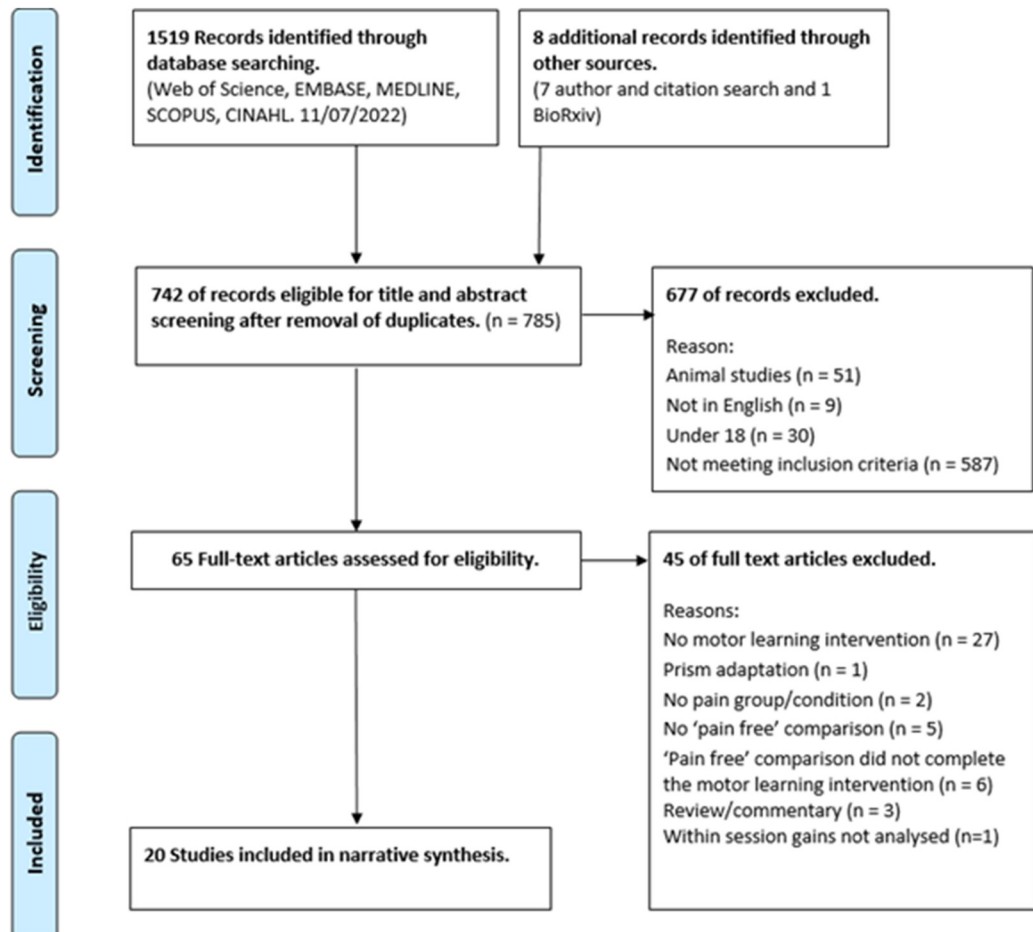

**Fig 1. PRISMA flow diagram.**

## 3.2. Risk of bias assessment

The ROB2 and ROBIN-I summaries for each outcome measure for each study can be seen in Tables 4–7. The agreement between reviewers when classifying ROB2 and ROBIN-I domains for all outcome measures was moderate, k = .582 (95% CI, .477 to .685), p < .000 and k = .642 (95% CI, .452 to .832), p < .000, respectively. Disagreements were discussed and a consensus was reached. The proportions of the ROB2 classifications for all 42 outcome measures included in the RCTs (experimental pain only) is shown in Fig 2.

Two outcome measures, accuracy at acquisition and retention, were classified as high risk of bias. The high-risk classifications were for the 'measurement of outcome' domain, related to inherent issues with ceiling effects or high baseline scores of accuracy measures during motor sequence learning paradigms [30]. The other thirty-eight outcome measures were classified as 'some concerns'. No outcome measures were classified as low risk of bias, and this needs to be considered when interpretating the findings of this review. Three out of the five clinical studies assessed using ROBIN-I were classified as 'serious'. The 'serious' classifications were for the 'confounding variables' section, related to not controlling or capturing information on potential sensory and motor deficits, associated with the clinical population, that may directly influence engagement in the task. Measures of the amount of pain participants were experiencing during the motor learning sessions was also lacking in these studies. The other two clinical

**Table 2. Key characteristics of included studies listed in order of motor learning paradigm.**

| Study details | Sample | Group characteristics: sample size, (Age (years)±SD), (location and Pain score ±SD) | Motor learning | Pain paradigm | Timing of pain | Task performance measure | Activity dependent plasticity measure |
|---|---|---|---|---|---|---|---|
| **Motor Sequence Learning** | | | | | | | |
| **Bilodeau et al (2016) [31] Between-subject** | N = 45 healthy subjects (66% female). | 1) Control n = 15 (28.8±8.8). 2) Local pain n = 15 (27.4±7.1). (dorsal wrist, NRS = 4.5) 3) Remote pain n = 15 (M28.5±9.5). (lateral left leg NRS 3.9) | **Explicit Sequential Finger tapping task** -(4-1-3-2-4) - 10 blocks of 30secs. | *Tonic/thermal* | Acquisition. Not pre and post testing. | 1) Error rate: mean number of errors per completed sequence. 2) Speed: number of completed sequences per 30s. | N/A |
| **Brown et al (2022) [44] Between-subject** | N = 38 subjects (66% female) | 1) Control n = 21 (24.76±3.85) 2) CNP group n = 17 (24.47±3.69) current episode of neck pain > 3 months & NDI > 4%. | **Implicit and Explicit modified serial reaction task involving reaching to central and peripheral targets on a screen.** | *Clinical neck pain* | Pain scores collected prior to, and after testing not during. | 1) Average time reaching to targets. 2) Total hand path distance during reaching | N/A |
| **Dancey et al (2014) [30] Between-subject** | N = 24 healthy subjects (46% female). | 1) Vehicle control n = 12 (23.4±2.0) 2) Pain group n = 12 (24.5±6.6) (Lateral elbow, NRS 4–5) | **Implicit Repetitive typing task**—Random order 3 numbers, (e.g., 9, 7, 8, 7, 9, 8) - 20mins. | *Tonic/Capsaicin (0.075% Zostrix)* | Acquisition, pre-testing, and post testing. | 1) Accuracy: number of correct responses pressed divided by the total number of combinations presented. 2) Reaction time from number sequence presentation to key press (ms). | 1) Somatosensory evoked potentials (SEPs) peaks. |
| **Dancey et al (2016) [32] Between-subject** | N = 24 healthy subjects (54% female). | 1) Vehicle control n = 12 (22.8±2.0). 2) Pain n = 12 (20.8 ±3.3) (Lateral elbow, NRS 6) | **Implicit Repetitive typing task**—Random 8-letter sequences of 4 letters (e.g., Z, D, P, Z, F, P, D, D) - 15mins. | *Tonic/Capsaicin (0.075% Zostrix)* | Acquisition, pre-testing, and post testing. Not retention. | 1) Accuracy: number of correct responses pressed divided by the total number of combinations presented. 2) Reaction time: number sequence presentation to key press (ms). | 1) Somatosensory evoked potentials (SEPs) peaks. |
| **Visuomotor Learning** | | | | | | | |
| **Andrew et al (2018) [46] Between-subject** | N = 24 subjects (50% female). | 1) Control n = 12 (22.75 range 21–27) 2) SCNP n = 12 (23.0 range 20–28) (grade I–II on Von Korff chronic pain grade scale) | **Explicit Visuomotor tracing task**—trace 4 different sequences of sinusoidal waves with thumb. 10 mins. | *Clinical/ subclinical neck pain* | No reported pain during experiment or data collected. | Accuracy: mean distance from a perfect trace expressed as a percentage (100% = one dot away from perfect trace). | 1) Somatosensory evoked potentials (SEPs) peaks. |
| **Boudreau et al (2007) [10] Within-subject** | N = 9 healthy subjects (22% female). (24 SD = 1.1) | 1) Vehicle control n = 9 2) Pain n = 9 (Tongue, VAS 5.1 ±0.6) | **Explicit tongue protrusion task**–using tongue to apply pressure to force plate to keep cursor within a moving target. 15 mins. | *Tonic/Capsaicin (1% cream)* | *Acquisition* | 1) Accuracy: percentage of time spent within the target. | 1) Single-pulse TMS measures. |

(*Continued*)

**Table 2.** (Continued)

| Study details | Sample | Group characteristics: sample size, (Age (years)±SD), (location and Pain score ±SD) | Motor learning | Pain paradigm | Timing of pain | Task performance measure | Activity dependent plasticity measure |
|---|---|---|---|---|---|---|---|
| **Dancey et al (2016) [33] Between-subject** | N = 24 healthy subjects (58% female). | 1) Vehicle control n = 12 (22.8±2.0) 2) Pain group n = 12 (20.8±3.3) (Lateral elbow, NRS 3–4) | **Explicit Visuomotor tracing task**—trace 4 different sequences of sinusoidal waves with thumb. 15 mins. | *Tonic/Capsaicin (0.075% Zostrix)* | Acquisition, pre-testing, and post testing. Not retention. | 1) Accuracy: mean distance from a perfect trace expressed as a percentage (100% = one dot away from perfect trace). | 1) Somatosensory evoked potentials (SEPs) peaks. |
| **Dancey et al (2019) [16] Between-subject** | N = 24 healthy subjects (75% female). | 1) Vehicle control n = 12 (20.7±1.4) 2) Pain group n = 12 (19.9±0.9) (Lateral elbow, NRS 4–5) | **Explicit Visuomotor tracing task**—trace 4 different sequences of sinusoidal waves with thumb. 15 mins. | *Tonic/Capsaicin (0.075% Zostrix)* | Acquisition, pre and post-test. Not retention. | Accuracy: mean distance from a perfect trace expressed as a percentage (100% = one dot away from perfect trace). | 1) Single-pulse TMS measures. |
| **Mavromatis et al (2017) [34] Between-subject** | N = 30 healthy subjects (50% female) | 1) Control n = 15 (27±6) 2) Pain group n = 15 (26±6) (Lateral border first metacarpal, NRS 3.5–4.5) | **Explicit modified version of the sequential pinch test**—10 blocks of 15 trials. | *Tonic/Capsaicin (1% cream)* | Acquisition | 1) Accuracy: Proportion of missed targets within a block. 2) Movement time: mean duration of sequences in a given block (secs) 3) Skill Measure: speed-accuracy trade-off equation. | 1) Single-pulse TMS measures. 2) Intracortical paired-pulse TMS measures. |
| **Rittig-Rasmussen et al (2014) [35] Between-subject** | N = 40 healthy subjects (60% female) aged 20–32 (mean = SD: 23 ±2) years | 1) Vehicle control n = 20 (isotonic saline, NRS 1.1±0.2) 2) Pain group n = 20 (right side of neck, NRS 4.8±0.4) | **Explicit shoulder elevation and depression training following a trace**—70 reps. | *Tonic/ Hypertonic saline injection* | Acquisition | 1) Error: deviations from the feedback curve between first five and last five reps (% improvement). | N/A |
| **Ballistic movements** | | | | | | | |
| **Ingham et al (2011) [13] Within-subject** | N = 9 healthy subjects (66% female). (M = 21.4 SD = 2.3) | 1) Vehicle control n = 9 (isotonic saline, VAS 0.2±0.4) 2) Local Pain group n = 9 (FDI, VAS 1.7±1.0) 3) Remote pain condition n = 9 (infra-patella fat pad, VAS 2.1±1.6) | **Explicit repeated voluntary finger movement opposing direction to that induced by TMS stimulation**—3 blocks of 8 sets of 50secs duration. | *Tonic/ Hypertonic saline injection (5% NaCl)* | Acquisition | 1) Acceleration of index finger during training. | 1) Single-pulse TMS measures. 2) TMS evoked peak acceleration. |
| **Parker et al (2017) [42] Between-subject** | N = 43 subjects (72% female). | 1) Control (no pain) n = 20 (71±7) 2) Hand OA n = 23 (72±6) (NRS >3 at least every other day) | **Explicit repeated voluntary finger movement opposing direction to that induced by TMS stimulation**– 30mins training, speed set by auditory cue. | *Clinical/ thumb osteoarthritis* | No reported pain during experiment or data collected. | 1) Percentage of accurate twitches: number of training twitches completed within 500ms of auditory cue within 27.5degrees of the training direction. | 1) Number of TMS-induced twitches in the baseline direction and training direction. 2) Single-pulse TMS measures. 3) Intracortical paired-pulse TMS measures. |

*(Continued)*

**Table 2.** (Continued)

| Study details | Sample | Group characteristics: sample size, (Age (years)±SD), (location and Pain score ±SD) | Motor learning | Pain paradigm | Timing of pain | Task performance measure | Activity dependent plasticity measure |
|---|---|---|---|---|---|---|---|
| **Vallence et al (2013)** [43] **Between-subject** | N = 29 subjects (59% female). | 1) Control (no pain) n = 18 (28±8) 2) CTTH n = 11 (35 ±13.2) (NRS 3.5±1.7) | **Explicit repeated voluntary thumb abduction movement**– rate 0.25Hz– 2 blocks of 225reps. | *Clinical/chronic tension type headache* | Pain score collected prior to training only. | 1) Peak acceleration of initial movement. | 1) Single-pulse TMS measures. |
| **Motor Adaptation** | | | | | | | |
| **Bouffard et al (2014)** [39] **Between-subject** | N = 30 healthy subjects (50% female). | 1) Control n = 15 (26±2.1) 2) Pain group n = 15 (26±1.4) (Ankle, VAS 4.8– 3.9) | **Explicit locomotor force adaptation task**–AFO applied a force field resisting right ankle dorsiflexion during mid-swing - 20mins walking (5 mins involved adaptation) | *Tonic/Capsaicin (1% cream)* | Acquisition not retention | 1) Mean absolute error of plantar flexion from a constructed baseline ankle angular displacement curve. 2) Peak plantar flexion error: direction the force pushes the foot. 3) EMG TA activity. | N/A |
| **Bouffard et al (2016)** [38] **Between-subject** | N = 37 healthy subjects (49% female). | 1) Control n = 24 (25.8±0.85) 2) Pain group n = 13 (26±1.15) (Ankle, VAS 5.5– 5.6) | **Explicit locomotor force adaptation task**–AFO applied a force field resisting right ankle dorsiflexion during mid-swing - 20mins walking (5 mins adaptation) | *Tonic/Capsaicin (1% cream)* | *Acquisition and retention* | 1) Mean absolute error of plantar flexion from a constructed ankle angular displacement curve measured at baseline. 2) Relative timing of error 3) EMG TA and Soleus activity. | N/A |
| **Bouffard et al (2018)** [40] **Between-subject** | N = 47 healthy subjects (45% female). | 1) Control n = 30 (25±1) 2) Pain group n = 17 (25±1) (Ankle, peak VAS 5.3) | **Explicit locomotor force adaptation task**–AFO applied a force field resisting right ankle dorsiflexion during mid-swing– 20mins walking (5 mins adaptation) | *Tonic/ Hypertonic saline injection (5% NaCl)* | Acquisition not retention | 1) Mean absolute error of plantar flexion from a constructed ankle angular displacement curve measured at baseline. 2) Relative timing of error 3) EMG TA and Soleus activity. | N/A |
| **Dupuis et al (2022)** [45] **Between-subject** | N = 17 subjects with isolated ankle pathology: Ankle fracture or OA (65% female) | 1) No pain group n = 9 (43.6±14.6) 2) Pain group n = 8 (54.9±13.9) (VAS Day 1 2.1±1.3, Day 2 2.3±0.8) | **Explicit locomotor force adaptation task**–AFO applied a force field resisting right ankle dorsiflexion during mid-swing– 20mins walking (5 mins adaptation) | *Isolated ankle pathology–self-reported constant clinical pain during learning task* | *Acquisition and retention* | 1) Mean absolute error of plantar flexion from a constructed ankle angular displacement curve measured at baseline. 2) Relative timing of error. 3) EMG TA activity. | N/A |
| **Lamothe et al (2014)** [36] **Between-subject** | N = 29 healthy subjects (52% female). | 1) Control n = 14 (26.6±4.8) 2) Pain group n = 15 (25.8±4.1) (Upper arm, VAS 7.8–7.5) | **Explicit ballistic reaching force adaptation task**–robotic exoskeleton– 100 reps per session | *Tonic/Capsaicin (1% cream)* | Acquisition not retention | 1) Final error (fERR). 2) The initial angle of deviation (iANG). | N/A |

(*Continued*)

**Table 2.** (Continued)

| Study details | Sample | Group characteristics: sample size, (Age (years)±SD), (location and Pain score ±SD) | Motor learning | Pain paradigm | Timing of pain | Task performance measure | Activity dependent plasticity measure |
|---|---|---|---|---|---|---|---|
| **Salomoni et al (2019) [37] Between-subject** | N = 22 healthy subjects (54% female). (M 28±6yr) | 1) Vehicle control n = 11 (isotonic saline) 2) Pain group n = 11 (deltoid, VAS 3–4.2) | **Explicit ballistic reaching force adaptation task**–robotic manipulandum– 100 reps per session | *Tonic/ Hypertonic saline (5% NaCl) injection* | Acquisition not retention | 1) Movement accuracy: peak hand speed, peak perpendicular error, and force adaptation index. 2) Initial rate of learning. 3) Movement strategy: measured using EMG. | N/A |
| **Ecological** | | | | | | | |
| **Arieh et al (2021) [41] Between-subject** | N = 30 healthy subjects (0% female). (Range 18–25) | 1) Control n = 10 2) Local Pain group n = 10 (lateral elbow, VAS 7.21±0.12) 3) Remote pain condition n = 10 (upper part of knee, VAS 7.13±0.13) | **Explicit Dart throwing Task**– 10 blocks of 15 dart throws. | *Tonic/Capsaicin (1% cream)* | Acquisition not pre and post-test. | 1) Throwing accuracy 2) Movement variability: motion camera analysis. | N/A |

Abbreviations: TMS, Transcranial Magnetic Stimulation; SEPs, Somatosensory Evoked Potentials; EMG, Electromyography; AFO, Ankle Foot Orthosis; FDI, First Dorsal Interosseous; PFC, Peak Force Command; TA, Tibilais Anterior; iANG, Initial Angle of Deviation; fERR, Final Error; CTTH, Chronic Tension Type Headache; SCNP, Subclinical Neck Pain; CNP, Clinical Neck Pain; OA, Osteoarthritis; NRS, Numerical Rating Scale; NDI, Neck Disability Index; VAS, Visual Analogue Scale.

studies were judged to have a 'moderate' risk of bias. Outcomes classified as high risk of bias using ROB2, or 'serious' or 'critical' risk of bias using ROBIN-I were omitted from the narrative synthesis and subsequent discussion.

### 3.3. Impact of pain on outcome measures

This analysis will discuss the impact of experimental and clinical pain paradigms separately for the five identified motor learning paradigm for the two primary outcome measures: task performance or activity dependent plasticity measure. A summary of findings for each study can be found in Table 3. Pain paradigms employed, specific details of outcome measures and GRADE scores are discussed for each motor learning paradigm. Tables 4–7 presents the studies findings, risk of bias judgement and GRADE score for each outcome measure (also see S1 and S2 Figs). Unless stated, risk of bias judgements for outcome measures were categorised 'some concerns.' Details on scoring of subsections for GRADE score can be found in S1 Table.

### 3.4. Impact of experimental pain

**3.4.1. Motor sequence learning.** Three studies [30–32] explored the impact of experimental pain on motor sequence learning. Implicit and explicit learning paradigms were explored across the studies. All three studies used a tonic experimental pain paradigm (two capsaicin and one thermal).

*3.4.1.1. Task performance measure.* Three outcome measures, accuracy, speed and reaction times, were employed by these studies. Accuracy, measured by number of errors, was classified

**Table 3. Summary of findings for each study for task performance and activity dependent plasticity measures.**

| Study author and date | Motor learning | Pain paradigm | Acquisition—task performance | Retention—task performance | Activity dependent plasticity |
|---|---|---|---|---|---|
| Bilodeau et al (2016) [31] | **Explicit Motor sequence learning** | **Tonic pain** | No significant difference in change in error rate or speed between groups. Overall error rates were low and as a result no improvement over time was seen. Speed did change over time. | No difference in change in error rate or speed between groups 24hrs after training. | N/A |
| Brown et al (2022) [44] | **Implicit and Explicit motor sequence learning** | **Clinical Pain** | Both groups demonstrated a significant decrease in time to target but no change in hand path distance across the explicit motor training. There was no significant difference between two groups at any time point. A significant decrease in time to target in the CNP group but not control group and significantly less hand path distance was observed in the control group but not the CNP group across implicit motor training. Comparison between groups demonstrated the control group had a significantly faster time to target at multiple time points during implicit motor learning. No analysis between groups across training available. | Both groups demonstrated a significant decrease in time to target but no change in hand path distance 30 mins after explicit motor training. No significant change in time to target or hand path distance was observed for either group 30 minutes after implicit motor training. | N/A |
| Dancey et al (2014) [30] | **Implicit Motor sequence learning** | **Tonic pain** | Accuracy improved across training in pain group. Accuracy in vehicle control group was high pre training and may explain reduced performance with training and underlie significant differences between groups. No significant difference in change in reaction times between groups. | N/A | N30 SEP peak significant increase in the pain group (increase 20.0%) but not in the control group (increase 9.0%) across training. No significant differences between group was seen in N20, N24, P25 or N18 SEP peak amplitudes. |
| Dancey et al (2016) [32] | **Implicit Motor sequence learning** | **Tonic pain** | Significant change in accuracy of vehicle control group only compared to baseline. Significantly higher accuracy levels in pain group at baseline may have influenced potential for improvement during training. No significant difference in change in reaction times between groups across training. | Significant change in accuracy of vehicle control group only, compared to baseline (see comment on acquisition). No significant difference in change reaction times from post learning to 48hrs (consolidation) between groups. | N20 SEP peak significantly changed in the placebo group (increase 35.5%) but not the pain group (Increase 11.2%) across training.<br><br>No significant differences between group was seen in N18, N30, P25 and N24 SEP peak amplitudes. |
| Andrew et al (2018) [46] | **Visuomotor learning** | **Clinical pain** | No significant difference in change in accuracy between groups across training. | Significantly better performance in the control group compared to the pain group at retention normalised to baseline. | N18 SEP peak significantly greater increase in the pain group (21.1%) compared to the control group (9.2%) across training. Significant difference in change in N24 SEP peak between groups across training. The control group decreased by 28.4% compared a 5.3% increase in pain group. No group differences N30 or N20. |

*(Continued)*

**Table 3.** (Continued)

| Study author and date | Motor learning | Pain paradigm | Acquisition—task performance | Retention—task performance | Activity dependent plasticity |
|---|---|---|---|---|---|
| Boudreau et al (2007) [10] | **Visuomotor learning** | **Tonic pain** | Improvement in accuracy of the task across training was significantly less in the pain condition than the vehicle control condition. | N/A | Significant decrease in M1 excitability in vehicle control condition but not pain condition across training. A significant increase in MEP values for 1.4T and 1.5T TMS intensity levels in vehicle group across training but no significant changes at any intensity in pain group. |
| Dancey et al (2016) [33] | **Visuomotor learning** | **Tonic pain** | No significant difference in change in accuracy from baseline between groups. Pain group outperformed control group before and after training. | Significant difference in change in accuracy from baseline to retention between groups (control group decreasing 70.5%, pain group decreased 46.0%) | A significantly difference in change between groups across training was seen in; N18 SEP peak (control group increase 1.7%, pain group decrease 18.5%), N24 SEP peak change in (control group decreasing 28.9%, pain group increase 3.0%) and N20 SEP peak (control group increased 48.9%, pain group decrease by 11.5%). No differences observed in N30 or P25 with training |
| Dancey et al (2019) [16] | **Visuomotor learning** | **Tonic pain** | No significant difference in change in accuracy from baseline between groups (Control decreased 48.7%, pain group decreased 35.2%) Pain group was significantly more accurate at baseline and post-acquisition. | No significant difference in change in accuracy from baseline between groups (Control decreased 21.9%, pain group decreased 10.7%). Pain group was significantly more accurate than control group. | Slope of TMS IO curves showed a significant increase in the control group compared to a non-significant decrease in the pain group. Neither group demonstrated a significant change in slope of TMS IO curves across training. |
| Mavromatis et al (2017) [34] | **Visuomotor learning** | **Tonic pain** | No significant difference in change in accuracy, movement time or speed-accuracy measure between points between groups. Throughout training pain group were significantly more accurate ($n^2$ = 0.284) resulting in better speed-accuracy performance measure. | N/A | The control group demonstrated significantly greater cortical excitability at mid training than the pain group. This difference was not observed at the end of training as the control group excitability had returned to baseline. No effect of group on SICI values. |
| Rittig-Rasmussen et al (2014) [35] | **Visuomotor learning** | **Tonic pain** | No significant difference in % error improvement between groups. | N/A | Cortical excitability measured but no analysis directly comparing across groups included. |
| Ingham et al (2011) [13] | **Ballistic movements** | **Tonic pain** | No difference in rate of improvement of finger acceleration between groups. | N/A | No difference in the change in TMS evoked peak acceleration between control and local pain group across training. In contrast to the local and control groups the remote group showed no change in TMS evoked peak acceleration across training. No difference in MEP amplitude or latency between groups or across training. |
| Parker et al (2017) [42] | **Ballistic movements** | **Clinical pain** | Significantly greater change in accuracy in the arthritis group (18.5%±25%) compared to the control group (0% ±46%). The control group was 10% more accurate than the pain group in the first 10% of trials and did not demonstrate group improvement across training. | N/A | The number of twitches in the baseline and training direction was not different across groups. Significantly greater SICF1.4 in arthritis group compared to control group. Significantly less SICI80 in the arthritis group compared to the control group. |

(*Continued*)

**Table 3.** (Continued)

| Study author and date | Motor learning | Pain paradigm | Acquisition—task performance | Retention—task performance | Activity dependent plasticity |
|---|---|---|---|---|---|
| Vallence et al (2013) [43] | **Ballistic movements** | **Clinical pain** | Significantly less learning in the CTTH group compared to the control group demonstrated by less acceleration change. | N/A | Significant increase in MEP amplitude across training in control group but not CTTH group. In the control group MEP amplitude was significantly increased at 10 and 20mins post but not at zero and five mins post. The returned to baseline by 30mins. |
| Bouffard et al (2014) [39] | **Motor Adaptation** | **Tonic pain** | No difference in change in mean absolute error, peak plantarflexion error or TA activity between the groups. | Control group demonstrated significantly lower mean absolute error than pain group suggesting impaired retention in the pain group. No difference in peak plantarflexion. | N/A |
| Bouffard et al (2016) [38] | **Motor Adaptation** | **Tonic pain** | No difference in change in mean absolute error or EMG activity between the groups across training. Significant between group differences in relative timing of error suggests pain group used less anticipatory strategies than control group. | No difference in change in mean absolute error between the groups. | N/A |
| Bouffard et al (2018) [40] | **Motor Adaptation** | **Tonic pain** | No difference in change in mean absolute error and EMG activity after PFC between the groups. Significant between group differences in relative timing of error and TA EMG activity before PFC on day one suggests pain group used less anticipatory strategies than control group. | No difference in change in mean absolute error across days between groups. No difference in timing errors between groups on day two suggests in the absence of pain on day two the pain group demonstrated anticipatory strategy. | N/A |
| Dupuis et al (2022) [45] | **Motor Adaptation** | **Clinical pain** | A significant difference was observed between early and late mean absolute error on day 1. No difference observed between groups. No effect of time or group observed for timing of errors or TA activity. | A significant difference in mean absolute error was observed between day 1 and day 2 but no difference in changes between groups. No group differences for timing of errors between days was observed. Pain group significantly decreased its TA activity prior to PFC compared to the control group, but no difference was observed in TA activity after PFC between groups. | N/A |
| Lamothe et al (2014) [36] | **Motor Adaptation** | **Tonic pain** | No significant group differences for iANG or fERR across training. The pain group made larger feedforward adjustments in anticipation of the force field perturbations. | No significant group differences in iANG or fERR at retention both retaining improvements made on day 1. | N/A |
| Study author and date | Motor learning | Pain paradigm | Acquisition—task performance | Retention—task performance | Activity dependent plasticity |
| Salomoni et al (2019) [37] | **Motor Adaptation** | **Tonic pain** | No significant difference in movement accuracy between groups across training. The control group adapted significantly quicker to force than the pain group during initial stages of learning. Muscle activity was significantly lower in the pain group during first exposure to the force field. | No difference in the capacity to compensate for perturbation. Difference in muscle activity was maintained even in the absence of pain at retention. | N/A |

(*Continued*)

**Table 3.** (Continued)

| Arieh et al (2021) [41] | **Ecological** | **Tonic pain** | No significant difference in change in throwing dart accuracy between groups across training. Significant greater coordination variability at elbow and wrist in pain groups compared to control group during deceleration phase. Significantly greater degree of wrist movement variability in local pain group compared to remote pain group during acceleration and compared to the control group in deceleration. | No significant difference in change in dart throwing accuracy between groups at retention (24hrs or 1 week). Above differences in movement variability during deceleration phase continued at retention (24hrs and 1 week) | N/A |
|---|---|---|---|---|---|

Abbreviations: TMS, Transcranial Magnetic Stimulation; SEPs, Somatosensory Evoked Potentials; MEP, Motor evoked potential; EMG, Electromyography; M1, Primary Motor cortex; SICI, Short Intracortical Inhibition; SICF, Short Intracortical Facilitation; iANG, Initial Angle of Deviation; fERR, Final Error; PFC, Peak Force Command; TA, Tibialis Anterior; CTTH, Chronic Tension Type Headache; SCNP, Subclinical Neck Pain; CNP, Clinical Neck Pain; OA, Osteoarthritis.

as high risk of bias, and therefore excluded. No impact of experimental pain on improvements in speed or reaction time during early stage/acquisition was reported. Two of the studies [31, 32] explored the impact of experimental pain during acquisition on a one-off performance of the motor sequence learning task 24hrs or 48hrs later. No impact of experimental pain was observed. The GRADE assessment for both outcome measures was judged to be low, suggesting limited confidence that experimental pain does not impact on the speed or reaction time of a motor sequence learning task at acquisition or retention.

*3.4.1.2. Activity dependent plasticity measures.* Two studies [30, 32] explored the impact of experimental pain on measures of neuroplasticity during motor sequence learning. Both explored the impact of capsaicin on SEPs using EEG. Only SEPs associated with changes across learning are discussed here. For detailed information on all SEPs examined in these studies please see Table 3. Both studies reported no impact of experimental pain on N24 and N18 SEPs associated with the cerebellum. Dancey et al. [30] reported a significantly greater increase in N30 SEP, associated with somatosensory integration and the motor cortex, across learning in the presence of experimental pain. Whereas, Dancey et al. [32] showed a significantly greater increase in N20 SEP, associated with the primary somatosensory cortex, following motor learning in the absence of experimental pain. GRADE scores for all SEP outcome measures in these studies were considered 'low', recommending limited confidence in the above impact on SEPs associated with the cerebellum, somatosensory integration and motor and primary somatosensory cortex.

**3.4.2. Visuomotor learning.** Five studies explored the impact of experimental pain on visuomotor learning. All five studies [10, 16, 33–35] used a tonic pain paradigm and explicit learning paradigms. One of the five experimental pain paradigms used intramuscular hypertonic saline injections [35] whereas the others used cutaneous capsaicin application.

*3.4.2.1. Task performance measure.* The impact of experimental pain was explored using four outcome measures: accuracy, number of errors, movement time and an accuracy speed trade off measure. Four out of the five studies [16, 33–35] demonstrated no impact of experimental pain on changes in accuracy following early-stage learning, although three studies [16, 33, 34] reported overall accuracy in the pain groups was higher at all data collection points. One study [10] reported experimental pain resulted in an impairment of learning across a single training session. This study utilised a tongue protrusion task whereas all the others utilised a task involving the upper limb. Experimental pain did not impact on any other outcome measure regardless of the method of application of the pain. Two studies [16, 33] explored the

**Table 4. Synthesis of evidence, risk of bias judgements and quality of evidence scores for the impact of tonic pain on individual task performance outcome measures.**

| Outcome measure | Studies | Motor learning paradigm | Tonic Pain paradigm | D1 | D2 | D3 | D4 | D5 | Overall risk of bias | Impact of pain of learning | Quality of evidence |
|---|---|---|---|---|---|---|---|---|---|---|---|
| | | | *Acquisition Phase* | | | | | | | | |
| **Accuracy–number of errors.** | Bilodeau (2016) [31] | Sequential finger tapping task | Cutaneous | L | L | L | H | SC | High | **NC** | Very low |
| | Dancey (2014) [30] | | | L | SC | L | H | SC | High | **+** | |
| | Dancey (2016) [32] | | | L | SC | L | H | SC | High | **-** | |
| **Accuracy–temporal/ spatial error from actual trace** | Boudreau (2007) [10] | Visuomotor task | Cutaneous | L | SC | L | L | SC | Some concerns | **-** | Low |
| | Dancey (2016) [33] | | | SC | L | L | SC | SC | Some concerns | **NC** | |
| | Dancey (2019) [16] | | | L | L | L | SC | SC | Some concerns | **NC** | |
| | Rittig-Rasmussen (2014) [35] | | Muscle | L | SC | L | L | SC | Some concerns | **NC** | |
| **Number of missed targets** | Mavromatis (2017) [34] | Visuomotor task | Cutaneous | L | L | L | L | SC | Some concerns | **NC** | Low |
| **Ecological measure** | Arieh (2021) [41] | Dart throwing | Cutaneous | SC | L | L | L | SC | Some concerns | **NC** | Low |
| **Movement error** | Bouffard (2014) [39] | Motor adaptation task | Cutaneous | L | L | L | L | SC | Some concerns | **NC** | Low |
| | Bouffard (2016) [38] | | | L | L | L | SC | SC | Some concerns | **NC** | |
| | Lamothe (2014) [36] | | | L | L | L | L | SC | Some concerns | **NC** | |
| | Bouffard (2018) [40] | | Muscle | SC | L | L | SC | SC | Some concerns | **NC** | |
| | Salomoni (2019) [37] | | | SC | L | L | L | SC | Some concerns | **NC** | |
| **Speed** | Bilodeau (2016) [31] | Sequential finger tapping task | Cutaneous | L | L | L | L | SC | Some concerns | **NC** | Low |
| **Movement time** | Mavromatis (2017) [34] | Visuomotor task | | L | L | L | L | SC | Some concerns | **NC** | Low |
| **Reaction times** | Dancey (2014) [30] | Sequential finger tapping task | Cutaneous | L | SC | L | SC | SC | Some concerns | **NC** | Low |
| | Dancey (2016) [32] | | | L | SC | L | SC | SC | Some concerns | **NC** | |
| **Accuracy/speed trade off** | Mavromatis (2017) [34] | Visuomotor task | Cutaneous | L | L | L | L | SC | Some concerns | **NC** | Low |
| **Timing of errors** | Bouffard (2016) [38] | Motor adaptation task | Cutaneous | L | L | L | SC | SC | Some concerns | **-** | Low |
| | Bouffard (2018) [40] | | Muscle | SC | L | L | SC | SC | Some concerns | **-** | |
| **Acceleration** | Ingham (2011) [13] | Repeated ballistic movements | Muscle | L | L | L | L | SC | Some concerns | **NC** | Low |
| | | | *Retention/consolidation* | | | | | | | | |
| **Accuracy–number of errors.** | Bilodeau (2016) [31] | Sequential finger tapping task | Cutaneous | L | L | L | H | SC | High | **NC** | Very Low |
| | Dancey (2016) [32] | | | L | SC | L | H | SC | High | **-** | |
| **Accuracy–temporal/ spatial error from actual trace** | Dancey (2016) [33] | Visuomotor task | Cutaneous | SC | L | L | SC | SC | Some concerns | **-** | Low |
| | Dancey (2019) [16] | | | L | SC | L | SC | SC | Some concerns | **NC** | |
| **Movement error** | Bouffard (2014) [39] | Motor adaptation task | Cutaneous | L | SC | L | L | SC | Some concerns | **-** | Low |
| | Bouffard (2016) [38] | | | L | SC | L | SC | SC | Some concerns | **NC** | |
| | Lamothe (2014) [36] | | | L | SC | L | L | SC | Some concerns | **NC** | |
| | Bouffard (2018) [40] | | Muscle | SC | SC | L | SC | SC | Some concerns | **NC** | |
| | Salomoni (2019) [37] | | | SC | SC | L | L | SC | Some concerns | **NC** | |
| **Speed** | Bilodeau (2016) [31] | Sequential finger tapping task | Cutaneous | L | L | L | L | SC | Some concerns | **NC** | Low |
| **Reaction times** | Dancey (2016) [32] | Sequential finger tapping task | Cutaneous | L | SC | L | SC | SC | Some concerns | **NC** | Low |
| **Relative timing of errors** | Bouffard (2016) [38] | Motor adaptation task | Cutaneous | L | SC | L | SC | SC | Some concerns | **NC** | Low |
| | Bouffard (2018) [40] | | Muscle | SC | SC | L | SC | SC | Some concerns | **NC** | |

Abbreviations: NC, No change in task performance; -, Pain caused a reduction in task performance; +, Pain caused an increase in task performance; Risk of bias: L = Low, SC = Some concerns, H = High.

**Table 5. Synthesis of evidence, risk of bias judgements and quality of evidence scores for the impact of tonic pain on individual activity dependent plasticity outcome.**

| Outcome measure | Studies | Motor learning paradigm | Tonic pain paradigm | D1 | D2 | D3 | D4 | D5 | Overall risk of bias | Impact of pain | Quality of evidence |
|---|---|---|---|---|---|---|---|---|---|---|---|
| | | | | | | *Acquisition Phase* | | | | | |
| N18 SEP Peak | Dancey (2016) [33] | Visuomotor Task | Cutaneous | SC | L | L | SC | SC | Some concerns | - | Low |
| | Dancey (2014) [30] | Sequential finger tapping task | | L | SC | L | SC | SC | Some concerns | | |
| | Dancey (2016) [32] | | | L | SC | L | SC | SC | Some concerns | NC | |
| N24 SEP Peak | Dancey (2016) [33] | Visuomotor Task | Cutaneous | SC | L | L | SC | SC | Some concerns | + | Low |
| | Dancey (2014) [30] | Sequential finger tapping task | | L | SC | L | SC | SC | Some concerns | NC | |
| | Dancey (2016) [32] | | | L | SC | L | SC | SC | Some concerns | NC | |
| N30 SEP Peak | Dancey (2016) [33] | Visuomotor Task | Cutaneous | SC | L | L | SC | SC | Some concerns | NC | Low |
| | Dancey (2014) [30] | Sequential finger tapping task | | L | SC | L | SC | SC | Some concerns | + | |
| | Dancey (2016) [32] | | | L | SC | L | SC | SC | Some concerns | NC | |
| N20 SEP Peak | Dancey (2016) [33] | Visuomotor Task | Cutaneous | SC | L | L | SC | SC | Some concerns | - | Low |
| | Dancey (2014) [30] | Sequential finger tapping task | | L | SC | L | SC | SC | Some concerns | NC | |
| | Dancey (2016) [32] | | | L | SC | L | SC | SC | Some concerns | - | |
| P25 SEP Peak | Dancey (2016) [33] | Visuomotor Task | Cutaneous | SC | L | L | SC | SC | Some concerns | NC | Low |
| | Dancey (2014) [30] | Sequential finger tapping task | | L | SC | L | SC | SC | Some concerns | NC | |
| | Dancey (2016) [32] | | | L | SC | L | SC | SC | Some concerns | NC | |
| TMS induced finger acceleration | Ingham (2011) [13] | Repeated ballistic movements | Muscle | L | L | L | L | SC | Some concerns | NC | Low |
| Single pulse MEPs | Boudreau (2007) [10] | Visuomotor Task | Cutaneous | L | SC | L | L | SC | Some concerns | + | Low |
| | Mavromatis (2017) [34] | | | L | L | L | L | SC | Some concerns | - midsession | |
| | Ingham (2011) [13] | Repeated ballistic movements | Muscle | L | L | L | L | SC | Some concerns | NC | |
| SICI | Mavromatis (2017) [34] | Visuomotor Task | Cutaneous | L | L | L | L | SC | Some concerns | NC | Very low |
| TMS-MEP response curves | Boudreau (2007) [10] | Visuomotor Task | Cutaneous | L | SC | L | L | SC | Some concerns | - | Very low |
| Slope of TMS-MEP response curves | Dancey (2019) [16] | Visuomotor Task | Cutaneous | L | L | L | SC | SC | Some concerns | - | Very low |

Abbreviations: TMS, Transcranial Magnetic Stimulation; SEPs, Somatosensory Evoked Potentials; MEP, Motor evoked potential; SICI, Short Intracortical Inhibition; SICF, Short Intracortical Facilitation; NC, No change in task performance; -, Pain caused a reduction in task performance; +, Pain caused an increase in task performance; Risk of bias: L = Low, SC = Some concerns, H = High.

impact of experimental pain during acquisition on a one-off performance of the visuomotor learning task 24hrs to 48hrs later. Dancey et al. [33] reported a significant decrease in accuracy in the pain group. In contrast, Dancey et al. [16] reported no differences in accuracy between groups. GRADE scores for all outcome measures were considered low. Therefore, based on the five studies there is limited confidence in the conflicting evidence of the impact of experimental pain on accuracy measures, and the lack of impact of experimental pain on temporal measures during a visuomotor learning task.

*3.4.2.2. Activity dependent plasticity measures.* Four studies [10, 16, 33, 34] explored the impact of experimental pain on activity dependent plasticity measures. One study investigated SEPs, [33] three studies [16, 33, 34] collected data on single pulse TMS measures, and one study explored paired pulse TMS measure. The single study exploring SEPs reported a significant difference in evoked potentials associated with the cerebellum (N18 and N24) between

**Table 6. Synthesis of evidence, risk of bias judgements and quality of evidence scores for the impact of clinical pain on individual task performance outcome measures.**

| Outcome Measure | Studies | Motor learning paradigm | D1 | D2 | D3 | D4 | D5 | D6 | D7 | Overall Risk of bias | Impact of pain | Quality of evidence |
|---|---|---|---|---|---|---|---|---|---|---|---|---|
| *Acquisition Phase* | | | | | | | | | | | | |
| **Accuracy–temporal/spatial error from actual trace** | Andrew (2018) [46] | Visuomotor task | S | L | L | L | L | M | L | Serious | **NC** | Very low |
| **Accuracy** | Parker (2017) [42] | Repeated ballistic movements | S | L | L | L | L | M | L | Serious | **+** | Very low |
| **Acceleration** | Vallence (2013) [43] | Repeated ballistic movements | S | L | L | L | L | M | L | Serious | **-** | Very low |
| **Time to target** | Brown (2022) [44] | Explicit Motor sequence task | M | L | L | L | L | M | L | Moderate | **NC** | Low |
| **Time to target** | Brown (2022) [44] | Implicit Motor sequence task | M | L | L | L | L | M | L | Moderate | **NC** | Low |
| **Distance hand moved** | Brown (2022) [44] | Explicit Motor sequence task | M | L | L | L | L | M | L | Moderate | **NC** | Low |
| **Distance hand moved** | Brown (2022) [44] | Implicit Motor sequence task | M | L | L | L | L | M | L | Moderate | **NC** | Low |
| **Movement error** | Dupuis (2022) [45] | Motor Adaptation | M | L | L | L | L | M | L | Moderate | **NC** | Low |
| **Timing of errors** | Dupuis (2022) [45] | Motor Adaptation | M | L | L | L | L | M | L | Moderate | **NC** | Low |
| *Retention/consolidation* | | | | | | | | | | | | |
| **Accuracy–temporal/spatial error from actual trace** | Andrew (2018) [46] | Visuomotor task | S | L | L | L | L | M | L | Serious | **-** | Very low |
| **Time to target** | Brown (2022) [44] | Explicit Motor sequence task | M | L | L | L | L | M | L | Moderate | **NC** | Low |
| **Time to target** | Brown (2022) [44] | Implicit Motor sequence task | M | L | L | L | L | M | L | Moderate | **NC** | Low |
| **Distance hand moved** | Brown (2022) [44] | Explicit Motor sequence task | M | L | L | L | L | M | L | Moderate | **NC** | Low |
| **Distance hand moved** | Brown (2022) [44] | Implicit Motor sequence task | M | L | L | L | L | M | L | Moderate | **NC** | Low |
| **Movement error** | Dupuis (20220) [45] | Motor Adaptation | M | L | L | L | L | M | L | Moderate | **NC** | Low |
| **Timing of errors** | Dupuis (2022) [45] | Motor Adaptation | M | L | L | L | L | M | L | Moderate | **NC** | Low |

Abbreviations: NC, No change in task performance; -, Pain caused a reduction in task performance; +, Pain caused an increase in task performance; Risk of bias: L = Low, M = Medium S = Serious, C = Critical.

groups. A decrease in N18 SEP in the pain group compared to the control group and a significant decrease in N24 SEP in the control group compared to the pain group was reported. Three studies reported a significant difference in single-pulse measures between groups. One study [10] reported a decrease in cortical excitability in the control group but not the pain group across training. In contrast, two studies reported a significant increase in the control group, halfway through training [34] and immediately post training [16]. Mavromatis et al. [34] reported the increase in excitability observed halfway through training, in the control group, returned to baseline by the end of training where it was not significantly different from the pain group. The same research group reported no significant difference in short intracortical inhibition following training. GRADE scores were low for all activity dependent plasticity outcome measures implying low confidence in the single study's findings on the impact of experimental pain on SEPs associated with the cerebellum and the conflicting results of three studies on the impact of experimental pain on single pulse TMS measures.

**Table 7. Synthesis of evidence, risk of bias judgements and quality of evidence scores for the impact of clinical pain on individual activity dependent plasticity outcome measures.**

| Outcome Measure | Studies | Motor learning paradigm | D1 | D2 | D3 | D4 | D5 | D6 | D7 | Overall Risk of bias | Impact of pain on plasticity | Quality of evidence |
|---|---|---|---|---|---|---|---|---|---|---|---|---|
| *Acquisition phase* | | | | | | | | | | | | |
| N18 SEP Peak | Andrew (2018) [46] | Visuomotor Task | S | L | L | L | L | M | L | Serious | N18 SEP peak + | Very low |
| | | | | | | | | | | | N24 SEP peak + | |
| | | | | | | | | | | | N30 SEP peak NC | |
| | | | | | | | | | | | N20 SEP peak NC | |
| Single pulse MEPs | Vallence (2013) [43] | Repeated ballistic movements | S | L | L | L | L | M | L | Serious | + | Very low |
| SICI | Parker (2017) [42] | Repeated ballistic movements | S | L | L | L | L | M | L | Serious | - | Very low |
| SICF | Parker (2017) [42] | Repeated ballistic movements | S | L | L | L | L | M | L | Serious | + | Very low |
| LICI | Parker (2017) [42] | Repeated ballistic movements | S | L | L | L | L | M | L | Serious | NC | Very low |
| Direction TMS induced twitches | Parker (2017) [42] | Repeated ballistic movements | S | L | L | L | L | M | L | Serious | NC | Very low |

Abbreviations: TMS, Transcranial Magnetic Stimulation; SEPs, Somatosensory Evoked Potentials; MEP, Motor evoked potential; SICI, Short Intracortical Inhibition; SICF, Short Intracortical Facilitation; NC, No change in task performance; -, Pain caused a reduction in task performance; +, Pain caused an increase in task performance; Risk of bias: L = Low, M = Medium S = Serious, C = Critical.

**3.4.3. Repeated ballistic movements.** One study [13] investigated the impact of experimental pain on learning using explicit repeated ballistic movement training of the upper limb digits. Ingham et al. [13] employed hypertonic saline injections to induce pain.

*3.4.3.1. Task performance measure.* The experimental pain study by Ingham et al. [13] reported no impact of either local or remote pain on improvement in finger acceleration during early-stage learning. GRADE scores for this outcome measure were considered low and therefore there is limited confidence in the absence of impact of experimental pain on learning employing repeated ballistic movement tasks.

*3.4.3.2. Activity dependent plasticity measures.* Ingham et al. [13] investigated two single pulse TMS activity dependent measures, TMS evoked peak finger acceleration and flexor digitorum indicis MEP. No significant difference was observed between local pain and control

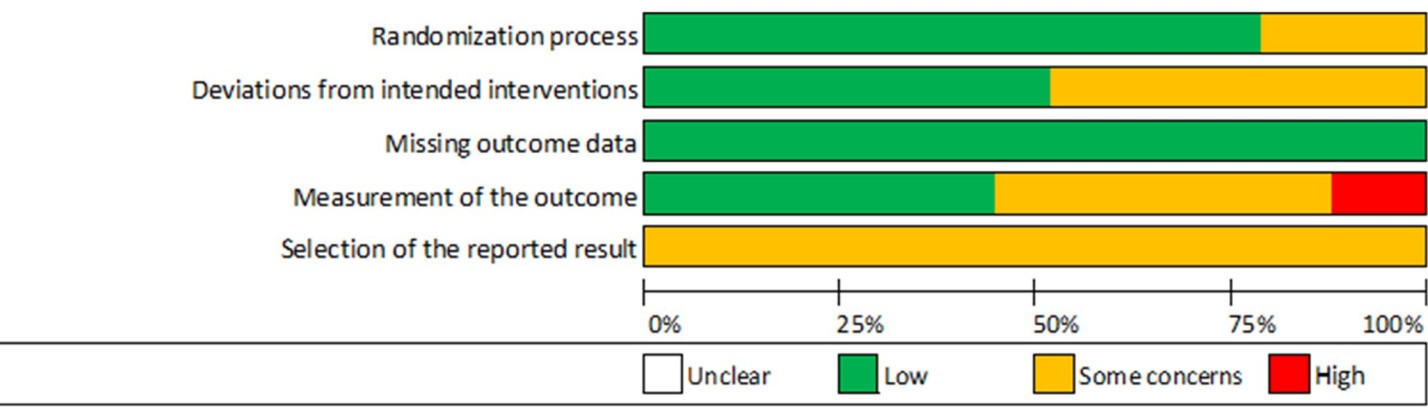

**Fig 2. The proportions of risk of bias classifications for 42 outcomes measures.**

groups for either measure. Only TMS evoked finger acceleration improved across training. In contrast, when experimental pain was applied remotely (patella fat pad) there was no change in TMS evoked finger acceleration following repeated ballistic finger training. Low GRADE scores suggest limited confidence in the absence of effect of pain on MEPs, and differing effect of local pain and remote pain on TMS finger induced acceleration.

**3.4.4. Motor adaptation.** Motor adaptation paradigms employed by the five studies exploring the impact of experimental pain on learning included: reaching [36, 37] and locomotion tasks [38–40]. All studies required participants to adapt to the application of an external force. Three studies [36, 38, 39] used capsaicin applied to cutaneous areas of relevant limb and two studies [37, 40] injected hypertonic saline into the muscle.

*3.4.4.1. Task performance measure*. Outcome measures assessed included movement error and timing of errors. All five studies reported no impact of experimental pain on movement errors during early stages of learning. One study [39] reported when pain was present during initial acquisition there was an impairment in subsequent learning during a subsequent training session (retention) 24hrs later. Four follow up studies [36–38, 40] from the same research group demonstrated no impact on retention. The same four studies [36–38, 40] reported a significant difference in movement strategies across learning. Bouffard et al. [38, 40] reported a difference in timing of errors suggesting reduced anticipatory strategies. This was supported by Salomoni et al. [37] who reported reduced muscle activity in the presence of experimental pain during first exposure to the force field. In contrast, Lamothe et al. [36] reported larger feedforward preparatory movements in the pain group. GRADE scores for all outcome measures were classified as low. Overall, there is limited confidence in the absence of impact of experimental pain on task performance measures during acquisition and retention or altered movement strategies in the presence of experimental pain during motor adaptation learning.

*3.4.4.2. Activity dependent plasticity measures*. No studies have explored the impact on experimental pain on activity dependent plasticity measures during motor adaptation learning tasks.

**3.4.5. Ecological learning.** A single study [41] utilised an ecological learning paradigm, dart throwing, to explore the impact of experimental pain on accuracy and movement variability.

*3.4.5.1. Task performance measure*. This study reported no impact of experimental pain on improvements in accuracy of performance, but increased movement variability at upper limb joints during the performance. Due to low sample size, GRADE score were considered low, resulting in limited confidence in findings that experimental pain does not impact on the accuracy of performance of ecological learning task.

*3.4.5.2. Activity dependent plasticity measures*. No studies have explored the impact of experimental pain on activity dependent plasticity measures during ecological learning tasks.

## 3.5. Impact of clinical pain

Five clinical pain studies [42–46] were included in this review. The three studies exploring the impact of; SCNP on a visuomotor learning task, [46] and OA of the thumb [42] and CTTH [43] on repeated ballistic movement tasks were judged to be at 'serious risk of bias' and therefore omitted from the analysis. The remaining two studies [44, 45] were judged to have 'moderate risk of bias' and their findings are discussed below. No studies explored the impact of clinical pain on ecological learning.

**3.5.1. Motor sequence learning.** One study [44] explored the impact of CNP on learning during a motor sequence learning task involving reaching for targets on a screen. In addition, the study's aims stated the authors wished to explore the different impact of clinical pain on implicit and explicit learning paradigms.

*3.5.1.1. Task performance measure.* Two outcome measures, 'time to target' and hand path distance, were collected at acquisition and retention following implicit and explicit motor learning blocks. The study found no impact of clinical neck pain on 'time to target' or hand path distance performance across implicit or explicit learning at either acquisition or retention. However, the authors reported the control group demonstrated significantly less time to reach the targets at multiple time points across learning, including baseline, compared to the clinical pain group during the implicit learning blocks only. GRADE scores were assessed as low for the above outcome measures. Therefore, there is limited confidence in the finding of an absence of impact of clinical pain on task performance measures during acquisition and retention during motor sequence learning.

*3.5.1.2. Activity dependent plasticity measures.* No studies have explored the impact of clinical pain on activity dependent plasticity measures during motor sequencing learning tasks.

**3.5.2. Motor adaptation.** A single study [45] explored the impact of pain from isolated ankle injuries on a motor adaptation locomotive task.

*3.5.2.1. Task performance measure.* Outcome measures assessed included movement error and timing of errors. This study found no impact of clinical pain on movement error or timing of errors at either acquisition or retention. Similar to findings for experimental pain, clinical pain was found to alter movement strategies associated with anticipation of movement during the motor learning. A reduction in TA muscle activity prior to the perturbation was observed, across the two days in the no pain group but not the pain group. GRADE scores for all outcome measures were classified as low. Overall, there is limited confidence in the absence of impact of clinical pain on task performance measures during acquisition and retention or altered movement strategies in the presence of clinical pain during motor adaptation learning.

*3.5.2.2. Activity dependent plasticity measures.* No studies have explored the impact on clinical pain on activity dependent plasticity measures during motor adaptation learning tasks.

# 4. Discussion

Our review question was 'does pain interfere with motor learning'? This narrative analysis highlights the array of different outcome measures, pain paradigms and motor learning paradigms utilised by the fifteen experimental pain studies and five clinical pain studies exploring this question. All outcome measures included within experimental pain and clinical pain studies have been judged to have low or very low quality, according to the guidelines set out by GRADE, suggesting limited confidence in the reported effect of experimental or clinical pain on measures of learning. This is primarily a result of the small-pooled sample sizes [47] lowering the score for imprecision, and the assigned risk of bias judgements.

## 4.1. Experimental pain

**4.1.1. Pain interference with task performance measures.** All fifteen studies used a tonic experimental pain paradigm. Nine out of 10 task performance measures explored during acquisition, across all five motor learning paradigms, demonstrated consistent results. No impact of pain on motor learning was found for all measures except for 'timing of errors' measured during motor adaptation. A smaller value for 'timing of errors' is associated with the participant switching to a more anticipatory strategy whereas a larger value suggests a more reactive strategy [38, 40]. The two studies exploring 'timing of errors' in motor adaptation reported that the presence of pain significantly reduced the transition to smaller values compared to the control group. This continued use of reactive strategies is supported by findings that EMG activity was lower in the pain group compared to the control group on first exposure to a force field [37]. This finding is consistent with previous reports of pain delaying

preparatory muscle activity during one-off tasks [48–50] but also extends the research to show an inability for participants to adapt this strategy despite practice.

Studies exploring the impact of pain on accuracy measures during acquisition of a visuo-motor task have reported contrasting results. One study demonstrated a reduction in motor learning across acquisition in the presence of pain compared to a control group [10]. In contrast, the other three studies [16, 33, 35] demonstrated no difference between groups across training, but two studies [16, 33] showed a significantly greater accuracy in the pain group at all points during the training.

Boudreau et al. [10] applied pain to the tongue during a tongue protrusion task compared to elbow pain during a motor tracing task involving the hand/wrist [16, 33]. It could therefore be argued that the interference of pain may vary depending on the location of pain. Rohel et al. [19] reported provisional evidence of a reduction in cortical excitability in response to hand and face pain but no reduction in forearm pain. The authors suggest that this may be due to greater sensory acuity and fine motor control of the hand and face associated with larger sensory and motor cortical representations.

A further difference between the studies is the type of task utilised. Complex or precision tasks such as the tongue protrusion task used by Boudreau et al. [10] may depend more on somatosensory feedback than tracing tasks. Pain has been shown to cause non-noxious sensory disturbances [51] and impaired processing of sensory information such as proprioception especially during complex tasks [52]. Researchers have theorised that somatosensory feedback plays a vital role in skill acquisition. Evidence from animal studies have demonstrated impaired learning of novel movements in the presence of lesions of the primary somatosensory cortex [53]. Similar findings have been demonstrated in humans using a repetitive TMS virtual somatosensory cortex lesion model [54]. In addition, peripheral manipulation of sensory information using lidocaine has been shown to decrease overall performance, increase errors and reduce reaction times during the same tongue protrusion task used by Boudreau and colleagues [14]. Both Vidoni et al. [54] and Boudreau et al. [14] utilised complex tasks requiring both visual information and graded movements during performance. Caution should be taken when comparing research using central and peripheral somatosensory disruptions, as mechanisms underlying central and peripheral sensorimotor interactions may differ [54].

An important consideration when interpreting the results from Boudreau et al. [10] is that application of the capsaicin depended on the participants engagement in the task. No data was collected on how the task was completed and therefore the impairment in learning could have resulted from reduced engagement in the task in attempt to reduce the painful stimuli. A challenge of future research in this area is to establish whether evidence of pain interference reflects pain attenuation of learning or pain disruption in performance during the learning task leading to reduced motor skill learning.

The findings by Dancey et al. [16, 33] that the pain group was more accurate during a visuo-motor tracing task of the wrist and hand at all time points compared to the control groups, suggests an interesting interaction between pain and task performance. It is possible that a pain stimulus delivered to the limb performing the motor task during motor learning increased attention to the limb, facilitating performance. Previous research, utilising other sensory inputs such as non-noxious electrical stimuli to the limb completing the learning task, reported increased attention facilitates neuroplasticity [55], measured using TMS. Whereas impairment in behavioural performance following motor learning occurs when instructions are given to intentionally divert attention away [56]. The latter is consistent with findings by Ingham et al. [13]. The authors reported decreased training induced changes in excitability related to hand motor learning in the presence of experimentally induced knee pain.

Attentional bias observed in people with chronic pain [57] may play a role in 'learning in pain', whereby, whether an individual attends to, or away from pain during motor learning, may determine whether improvements in performance are facilitated or impaired. The interaction between pain, attention and motor learning interference may also be influenced by cognitive constructs, such as fear and motivation [11]. The balance of 'motivation to protect' versus 'motivation to complete the task' may influence the resultant interference. An important question for future research design is 'what is the value of the pain experience to the participant?' and 'what factors influence the participants assessment of value, such as fear and catastrophizing?'.

Task performance measures recorded more than 24hrs after acquisition demonstrated conflicting results across studies. Definitions and approaches to measuring retention varies across the studies. Dancey et al. [32, 33] retested task performance more than 24hrs post acquisition, focusing mainly on improvements in the absence of further training (offline learning) [6]. In contrast, Bouffard et al. [38–40] Salomoni et al. [37] and Lamothe et al. [36] observed the changes in task performance across a further training session (online learning–long term retention) [58]. Physiological processes underlying the above two processes may vary and therefore comparison of findings is limited.

**4.1.2. Pain interference with activity-dependent plasticity measures.** There is a lack of consensus between studies on the impact of experimental tonic pain on activity dependent plasticity measures of motor learning, such as TMS single pulse measures and SEP peaks. Paired pulse TMS measures were limited to single studies and therefore any conclusions would be premature. No studies have explored activity dependent plasticity using functional MRI. The findings in this review are consistent with a systematic review by Stanisic et al. [23] who reported acute experimental pain may influence training induced neuroplasticity defined by TMS. The authors suggested the impact of pain on training induced neuroplasticity may be dependent on the motor learning paradigms utilised. The observed variation in the findings in the present review could be in part due to the respective tasks.

Paparella et al. [59] demonstrated selectivity of cortical, corticospinal and intra-cortical changes in response to different learning tasks. Ingham et al. [13] employed a simple ballistic thumb movement task compared to the precision tasks employed in the other two studies. It could be argued that the tasks used in these studies were of contrasting difficulty, and therefore with the addition of pain, the cognitive load may differ [52], which may influence the change in TMS acquired corticomotor excitability [34].

The demands of the task may explain some of the differences seen in the effects of pain on SEPs. Dancey et al. [32, 33] reported a significant increase in N20 SEP in the control group compared to the pain group. In contrast, Dancey et al. [30] reported no changes in the control group or between the two groups. This study from this research group utilised a pure motor typing task. SEPs reflect precognitive sensory processing and are markers of somatosensory integration. N20 SEP represents the earliest cortical processing in the somatosensory area [32]. The findings discussed above may suggest a role for the somatosensory area in processing of relevant sensory information associated with learning of a complex but not a simple motor task, and that this processing can be disrupted by sensory stimuli such as experimental pain. Further evidence of the importance of adapting to sensory stimuli was noted by Lamothe et al. [36] during motor adaptation task. They suggested the pain group utilised strategies to minimise the need for online adjustments in task performance.

## 4.2. Clinical pain

The two clinical pain studies [44, 45] included in this review investigated the impact of clinical pain on task performance measures only. Both studies included participants with chronic pain

only, defined as those experiencing symptoms for more than 3 months. No impact of clinical pain was reported on the four outcome measures during either acquisition or retention. This provides limited evidence that chronic pain effects on motor learning are similar to acute experimental pain effects discussed above. Although, caution must be taken when comparing the effects of acute and chronic pain. Research has suggested that both the global and local neural activity of chronic clinical pain states differs from acute pain states [60–64].

Brown et al. [44] reported clinical pain impeded 'time to target' during all learning blocks in implicit motor learning but not explicit learning. The authors hypothesise that participants with chronic pain may approach tasks with increased caution in the absence of explicit information pertaining to the task. Meulders et al. [65] labelled this concept as contextual pain related fear which results in sustained anticipatory anxiety resulting from unpredictability of pain (knowing the pain will occur but not knowing when exactly).

## 4.3. Possible mechanisms for pain interference of motor learning

Interference of motor learning has been shown to occur when two motor tasks compete for the same neural substrate [3, 66]. Lange et al. [67] points to inhibition of shared networks as a mechanism underlying interference with motor learning. It has been well documented that pain and movement activate similar areas of the central nervous system including spinal cord, cerebellum, basal ganglia, anterior cingulate cortex, premotor and primary motor cortex [60, 68–70]. Previous systematic reviews provide evidence that experimental pain decreases excitability of the motor pathway including the motor cortex [19, 21, 71]. These findings contrast with the findings of this study which found no consistent evidence that experimental pain inhibits cortical changes associated with motor learning. Sub-grouping analysis performed by Sanderson et al. [71] suggests that the impact of pain on cortical excitability may be dependent on the specific pain paradigm or tissue stimulated. The authors reported increased cortical excitability when hypertonic saline was injected into non-contractile tissue and a decrease when injected into contractile tissue. Only four studies out of the 15 experimental pain studies included in the present review induced pain in muscle tissue and therefore further research is needed to explore whether the physiological effects on motor learning are different from cutaneous induced pain models.

In agreement with this review, Parker et al. [72] and Sanderson et al. [71] report no consistent evidence that clinical pain alters motor cortical excitability measured through single pulse measures. In contrast, Parker et al. [72] reported a reduction in intra-cortical inhibition, a mechanism linked to GABAergic inhibition and potential modulator of motor learning, in chronic pain populations. The majority of studies included in this review utilised single pulse measures of cortical excitability and therefore it is not possible to come to any conclusions around the impact of clinical pain on disinhibition of cortical pathways following motor learning. It is possible that pain interference with motor learning may be specific to pain paradigms and/or sub-groups of a population that demonstrate disruption of the normal inhibitory neural networks within the motor pathways, such as neuropathic pain [72]. Further discussions regarding this are limited by the fact that both clinical pain populations employed by the two studies included in this review had an absence in symptoms associated with the above neural changes, assessed using pain pressure thresholds and sensory perception.

Timing of the interfering stimuli may also be a factor that influences whether resultant interference is observed. Previous evidence of interference has presented the interfering stimuli prior to, or immediately following the motor learning task [3, 66, 73], potentially drawing resources or attention away from the processes needed to acquire or consolidate the learning. For all studies included within this review participants were exposed to pain during the

training session. A number of these studies suggested that such timing potentially drew attention towards the task, increasing performance [16, 33]. Future studies could expose participants to acute experimental pain one to four hours post learning in order to investigate pain interference with the consolidation process.

## 5. Strengths and limitations

This is the first systematic review to provide an in-depth analysis of the impacts of both experimental pain and clinical pain on individual outcome measures for different motor learning paradigms. Employing this approach has attempted to minimise comparisons between motor learning paradigms which potentially have different underlying neural mechanisms [17], and outcome measures which measure efficiency and effectiveness versus those which explore performance strategies. Consequently, this review demonstrates, across different motor learning paradigms, efficiency and effectiveness of performance can significantly increase across learning despite the presence of pain and alternative motor strategies. This is contrary to the conclusions reported by a recent systematic review [22] but likely reflects the different approach outlined above. The limitations of this review need to be considered when interpreting the findings. Due to the heterogeneity of the motor learning paradigms, pain paradigms and outcome measures a meta-analysis was not used. Overall, there was limited confidence in the impact of pain on most outcome measures due in part to small sample sizes and the assigned level of risk of bias for each study. In an attempt to reduce heterogeneity of experimental design and competing physiological mechanisms, studies using interventions to lessen pain prior to motor learning and prism-adaptation paradigms were excluded. It is possible that omitting these studies may have rejected findings that may have contributed different perspectives to the research question.

### 5.1. Future research

To further our knowledge of pain interference with motor learning, both pain paradigms and motor learning paradigms need to be carefully considered. Future clinical pain studies may benefit from investigating chronic pain populations with evidence of neuroplastic changes, such as synaptic potentiation or reduced grey matter volumes. Subgrouping clinical pain populations based on their presentations or propensity for potentiation [74], and reporting on other sensory or psychological presentations that coexist with selected clinical pain conditions may provide interesting results. Experimental pain research may benefit from utilising pain paradigms that better reflect clinical pain. For example, movement related pain paradigms, recurrent pain paradigms, or paradigms with associated secondary hyperalgesia such as nerve growth factor injections.

The suggestion that interference may be task-dependent has been fuelled by reports neural correlates underpinning learning in different motor paradigms may vary. This review provides little evidence to support this conclusion. Future research studies exploring the impact of a single pain paradigm across different motor learning paradigms, employing more complex and ecologically relevant motor tasks, and including measures of attention, such as eye tracking, may provide interesting insights.

## 6. Conclusions

The present review concludes there is limited confidence in the reported effect of pain on motor learning. The majority of the research suggests with repeated task practice, individuals experiencing experimental tonic pain or clinical pain, will improve their task performance by the same amount as individuals who perform the same repeated task practice without pain. Although, this may depend on both the location and target tissue the pain is applied to and the

type of task. This finding fails to provide support that pain disrupts non-pain related goals or interferes with the motor learning process. This review utilises a narrative analysis to highlight the challenges in research design when exploring pain interference with motor learning and hopes to act as a guide to researchers regarding future directions and study design.

## Supporting information

**S1 Fig. Fig 3 Bubble chart displaying impact of pain on task performance across motor learning for spatial and temporal measures during acquisition and retention.** Outcome measures from 18 studies: 15 tonic experimental and five clinical pain.
(TIF)

**S2 Fig. Fig 4 Bubble chart displaying impact of pain on activity dependent measures across motor learning during acquisition.** Outcome measures from 18 studies: 15 tonic experimental and five clinical pain.
(TIF)

**S1 Table. Tables 8–10 Summary of GRADE judgements for all outcome measures.**
(DOCX)

**S1 File. Search strategy.**
(PDF)

**S1 Checklist. PRISMA 2020 checklist.**
(DOCX)

## Author Contributions

**Conceptualization:** David Matthews, Edith Elgueta Cancino, Deborah Falla, Ali Khatibi.

**Data curation:** David Matthews, Edith Elgueta Cancino, Deborah Falla, Ali Khatibi.

**Formal analysis:** David Matthews, Edith Elgueta Cancino, Deborah Falla, Ali Khatibi.

**Methodology:** David Matthews, Edith Elgueta Cancino, Deborah Falla, Ali Khatibi.

**Project administration:** David Matthews.

**Resources:** David Matthews, Edith Elgueta Cancino.

**Supervision:** Deborah Falla, Ali Khatibi.

**Validation:** David Matthews, Edith Elgueta Cancino, Deborah Falla, Ali Khatibi.

**Visualization:** David Matthews, Edith Elgueta Cancino, Deborah Falla, Ali Khatibi.

**Writing – original draft:** David Matthews, Edith Elgueta Cancino, Deborah Falla, Ali Khatibi.

**Writing – review & editing:** David Matthews, Edith Elgueta Cancino, Deborah Falla, Ali Khatibi.

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
