## [Decision Letter · Decision Letter 0]

24 Jun 2022

PONE-D-22-06510Exploring pain interference with motor skill learning in humans: A systematic reviewPLOS ONE

Dear Dr. Matthews,

Thank you for submitting your manuscript to PLOS ONE. After careful consideration, we feel that it has merit but does not fully meet PLOS ONE’s publication criteria as it currently stands. Therefore, we invite you to submit a revised version of the manuscript that addresses the points raised during the review process.

We look forward to receiving your revised manuscript.

Kind regards,

Yih-Kuen Jan, PhD

Academic Editor

PLOS ONE

Journal Requirements:

Reviewers' comments:

Reviewer's Responses to Questions

**Comments to the Author**

1. Is the manuscript technically sound, and do the data support the conclusions?

Reviewer #1: Yes

Reviewer #2: Yes

2. Has the statistical analysis been performed appropriately and rigorously? 

Reviewer #1: N/A

Reviewer #2: Yes

3. Have the authors made all data underlying the findings in their manuscript fully available?

Reviewer #1: Yes

Reviewer #2: Yes

4. Is the manuscript presented in an intelligible fashion and written in standard English?

Reviewer #1: Yes

Reviewer #2: Yes

5. Review Comments to the Author

Reviewer #1: The authors systematically reviewed evidence on the impact of experimental and clinical pain on motor learning. Based on 18 studies included in the review, there was limited evidence of task performance changes due to pain. The paper is well written and the approach is appropriate but a few questions remain.

Overall, how does the paper differ form recent work by Izadi et al., [1] examining motor learning in response to experimental pain models? Be sure to compare and contrast findings in discussion.

Methods.

2.2.4. Outcomes. Why were activity dependent plasticity measures limited to amplitude, temporal, or spatial characteristics? Why were frequency domain measures not considered?

2.2.5. Study design. Why were studies including treatments as an adjunct to motor learning excluded? Wouldn’t those studies provide significant utility in evaluating potential mechanisms by which learning could be improved in the pretense of pain?

2.3. The search should be updated to reflect new papers published in the past year.

Discussion.

Be sure to discuss how findings compare and contrast with recent reviews [1,2] and findings in clinical pain [3]. Discuss limitations of current approach and how current findings compare with recent work.

Literature Cited:

[1] Izadi M, Franklin S, Bellafiore M, Franklin DW. Motor Learning in Response to Different Experimental Pain Models Among Healthy Individuals: A Systematic Review. Front Hum Neurosci. 2022;16:863741. Published 2022 Mar 24. doi:10.3389/fnhum.2022.863741

[2] Parker, Rosalind S., et al. "Is motor cortical excitability altered in people with chronic pain? A systematic review and meta-analysis." Brain stimulation 9.4 (2016): 488-500.

[3] Brown, Michael R., Kirkwood E. Personius, and Jeanne Langan. "Participants with mildly-disabling chronic neck pain perform differently during explicit compared to implicit motor learning of a reaching task." PloS one 17.4 (2022): e0266508

Reviewer #2: The authors systematically reviewed the literature to explore the impact of pain on motor skill learning. However, the review suggests that there was no impact of pain on any task performance measures due to the limited evidence and heterogeneity.

MAJOR COMMENT

Introduction

1. Line 81-84: In my opinion, a brief introduction to the three pain paradigms and various motor paradigms might be needed to enable the readers to understand the differences.

Discussion

2. Line 404-406: I recommend the authors elaborate more on how somatosensory feedback affects task performance.

3. In my opinion, the experimental pain (especially the way to apply capsaicin) would not really damage the human tissue but other ways might cause some injuries, which might result in different physiological changes. The evident difference should be noted and discussed.

4. Line 415-416: It is interesting that the increased attention of participants caused by pain might facilitate task performance. Is that possible that other sensory inputs would cause similar results?

5. In the clinical pain studies, were they all acute pain or chronic pain? Acute pain (eg. acute ankle sprain) and chronic pain (eg. osteoarthritis) might influence motor learning a lot. Much evidence has pointed out that chronic pain would change the excitability level or activation area of the cerebral cortex. I recommend the authors could put the related findings in the discussion.

MINOR COMMENT

Table 1 and Table 2. Please provide the full term for abbreviations in the footnotes like Table 3.

6. PLOS authors have the option to publish the peer review history of their article (what does this mean?). If published, this will include your full peer review and any attached files.

Reviewer #1: No

Reviewer #2: No

---

## [Author Response · Author response to Decision Letter 0]

8 Aug 2022

Further details of response can be found in 'Response to reviewers' file uploaded.

Authors response: Changes made to title page and formatting of titles. File naming uploaded as per guidelines on PLOSone website. 

Authors response: All tables uploaded to manuscript. Supplementary figures and tables uploaded separately. Titles of supplementary material added to bottom of manuscript.

3. How does the paper differ from recent work by Izadi et al., [1] examining motor learning in response to experimental pain models? Be sure to compare and contrast findings in discussion. Be sure to discuss how findings compare and contrast with recent reviews [1,2] and findings in clinical pain [3]. Discuss limitations of current approach and how current findings compare with recent work.

Authors response: Two systematic reviews (Izadi et al. (2022) and Stanisic et al. (2022) have been published in this research area since the original submission of this manuscript. The below comments have been added to the introduction and discussion to reflect this and discuss the different methodological approaches and review findings. A discussion around the impacts of pain on cortical excitability has been adapted. The findings of the article mentioned by the reviewer [2] exploring the impact of chronic pain on cortical excitability (Parker et al 2016) has been considered when interpretating the results of this review. A section on limitations has been added to the discussion.

4. Why were activity dependent plasticity measures limited to amplitude, temporal, or spatial characteristics? Why were frequency domain measures not considered?

Authors response: Activity measures were not limited to amplitude, temporal or spatial measures and the method has been adjusted to reflect this. No frequency domains of SEPs were explored as a measure in any included study and therefore no further discussion on this was deemed appropriate. To clarify this a sentence was added to the results.

5. Why were studies including treatments as an adjunct to motor learning excluded? Wouldn’t those studies provide significant utility in evaluating potential mechanisms by which learning could be improved in the pretense of pain?

Authors response: Scoping searches prior to the preparation of the systematic review protocol revealed two studies including other treatments alongside motor learning tasks. These included; manipulation (Baarbe et al., 2018) and repeated peripheral magnetic neurostimulation (Massie-Alarie., 2017). The authors agreed including these articles would add to the existing diversity of the reviews methods which would impact on the ability for the review to provide coherent summary of research in this area for readers.

6. The search should be updated to reflect new papers published in the past year.

Authors response: Search updated to July 2022, see line 138. Two further studies found meeting the criteria and included in the analysis. Two systematic reviews exploring similar research questions were found and these were included in the intro and discussion (see point 3).

7. Line 81-84: In my opinion, a brief introduction to the three pain paradigms and various motor paradigms might be needed to enable the readers to understand the differences.

Authors response: Definitions of pain paradigms added to text and motor learning definitions included in new table (Table 1). Line 118.

8. Line 404-406: I recommend the authors elaborate more on how somatosensory feedback affects task performance.

Authors response: Further research exploring the impact of somatosensory feedback/processing is added to the discussion including changes at different levels of the sensory pathway.

9. In my opinion, the experimental pain (especially the way to apply capsaicin) would not really damage the human tissue but other ways might cause some injuries, which might result in different physiological changes. The evident difference should be noted and discussed.

Authors response: Capsaicin models were included in this study and were not considered to be a paradigm that may result in tissue damage. The method outlines those pain paradigms that were excluded due to be associated with potential changes in the muscle tissue such as DOMS (related to micro tears in the muscle and a healing process) and neurological disease (associated with reduced neural drive and associated muscle wasting). The authors agree clinical pain studies can be associated with tissue damage. This is highlighted during discussions around ROBIN-I, risk of bias and future studies. Musculoskeletal/neurological changes associated with the pain conditions could act as confounding variables. A lack of screening of clinical pain populations at baseline for the above confounding variables was a key determinant of the serious risk of bias judgements and subsequent exclusion of those studies from further analysis. The remaining two clinical pain studies included for further analysis did not differ in terms of baseline measures related to motor and sensory testing and therefore further discussions on this was not included.

10. Line 415-416: It is interesting that the increased attention of participants caused by pain might facilitate task performance. Is that possible that other sensory inputs would cause similar results?

Authors response: The following paragraph was adapted for clarity, the research referred to when exploring impact of attention on neuroplastic changes used non-noxious sensory stimuli or auditory commands. A further question posed at the end of this paragraph is whether the attentional changes are a result of the painful stimulation or any sensory information and the importance of choice of control groups to explore this.

11. In the clinical pain studies, were they all acute pain or chronic pain? Acute pain (eg. acute ankle sprain) and chronic pain (eg. osteoarthritis) might influence motor learning a lot. Much evidence has pointed out that chronic pain would change the excitability level or activation area of the cerebral cortex. I recommend the authors could put the related findings in the discussion.

Authors response: The two included clinical studies after quality assessment were both chronic pain paradigms and this has been added to the discussion. The studies found no effect of chronic pain on learning and therefore does not suggest differences between chronic and experimental acute pain effects on learning. A caveat to this, stated in the discussion, is sub-groups of chronic pain patients such as neuropathic pain may have different interactions with learning and chronic pain and acute pain mechanisms differ. The 3 clinical pain studies removed from the narrative review all explored different types of chronic pain with only one study demonstrating impact on motor learning. As mentioned previously as sensory and motor symptoms were not collated in these studies it is difficult to subgroup these studies. Worth noting one of these studies looked at osteoarthritis of the thumb and found no change in behaviour learning outcomes resulting from motor learning. In response to the comment stating chronic pain changes excitability of the cortex, two systematic reviews (including Parker et al 2016) are presented in the discussion which conclude that there is evidence of no impact of chronic pain on single pulse measures of cortical excitability but possibly changes in intracortical inhibition.

12. Table 1 and Table 2. Please provide the full term for abbreviations in the footnotes like Table 3

Authors response: Footnotes/abbreviations added to all tables.

---

## [Decision Letter · Decision Letter 1]

28 Aug 2022

Exploring pain interference with motor skill learning in humans: A systematic review

PONE-D-22-06510R1

Dear Dr. Matthews,

We’re pleased to inform you that your manuscript has been judged scientifically suitable for publication and will be formally accepted for publication once it meets all outstanding technical requirements.

Kind regards,

Yih-Kuen Jan, PhD

Academic Editor

PLOS ONE

Additional Editor Comments (optional):

Reviewers' comments:

Reviewer's Responses to Questions

**Comments to the Author**

1. If the authors have adequately addressed your comments raised in a previous round of review and you feel that this manuscript is now acceptable for publication, you may indicate that here to bypass the “Comments to the Author” section, enter your conflict of interest statement in the “Confidential to Editor” section, and submit your "Accept" recommendation.

Reviewer #1: All comments have been addressed

Reviewer #2: All comments have been addressed

2. Is the manuscript technically sound, and do the data support the conclusions?

Reviewer #1: Yes

Reviewer #2: Yes

3. Has the statistical analysis been performed appropriately and rigorously? 

Reviewer #1: Yes

Reviewer #2: Yes

4. Have the authors made all data underlying the findings in their manuscript fully available?

Reviewer #1: Yes

Reviewer #2: Yes

5. Is the manuscript presented in an intelligible fashion and written in standard English?

Reviewer #1: Yes

Reviewer #2: Yes

6. Review Comments to the Author

Reviewer #1: The authors have more than adequately addressed my comments. No further questions remain at this time.

Reviewer #2: (No Response)

7. PLOS authors have the option to publish the peer review history of their article (what does this mean?). If published, this will include your full peer review and any attached files.

Reviewer #1: No

Reviewer #2: No

---

## [Editor Report · Acceptance letter]

2 Sep 2022

PONE-D-22-06510R1 

Exploring pain interference with motor skill learning in humans: A systematic review 

Dear Dr. Matthews:

I'm pleased to inform you that your manuscript has been deemed suitable for publication in PLOS ONE. Congratulations! Your manuscript is now with our production department. 

Kind regards, 

on behalf of

Dr. Yih-Kuen Jan 

Academic Editor

PLOS ONE